# A large-scale MIMO antenna system for 5G IoT applications

Hung Tran-Huy[1]*, Thuy Nguyen Thi[2], Muhammad Aslam[3], Hong Nguyen Tuan[4], Tung The-Lam Nguyen[5]

1 Faculty of Electrical and Electronic Engineering, PHENIKAA University, Hanoi, Vietnam, 2 RnD Center, ESSYS Co., Ltd, Incheon, Korea, 3 Department of Artificial Intelligence, Sejong University, Seoul, Korea, 4 Center for High Technology Development, Vietnam Academy of Science and Technology (VAST), Ha Noi, Vietnam, 5 IT Department, FPT University, Greenwich Vietnam, Hanoi, Vietnam

* hung.tranhuy@phenikaa-uni.edu.vn

## Abstract

A multiple-input-multiple-output (MIMO) antenna system with low-profile and small element spacing characteristics is presented in this paper. This antenna contains multiple elements arranged in both E-plane and H-plane configurations. The original strong coupling between the MIMO elements can be suppressed by exciting orthogonal operating modes. To achieve this, a half-wavelength microstrip line and a quarter-wavelength grounded stub are utilized to decouple the H- and E-plane MIMO arrays. A 2 × 2 antenna prototype is fabricated and measured to demonstrate the decoupling concept's feasibility. The measured impedance bandwidth is from 4.78 to 4.81 GHz. Across this band, the isolation is better than 15 dB with extremely small edge-to-edge distances of 0.032λ and 0.026λ in the E- and H-plane, respectively. Featuring the simple decoupling structure, small element spacing, and the capability of extending to a large-scale 2 × N array, the proposed antenna can be used for 5G Internet of Things (IoT) applications operating at the N79 frequency band.

## Introduction

The development of wireless communication technology, including 5G/B5G, Wi-Fi 6E, and low-earth orbit (LEO) satellites, has significantly changed our lifestyles in recent decades. There has also been a substantial surge in the prevalence of the Internet of Things (IoT) devices. To support potential wireless applications, such as multimedia, intelligent transportation systems, and so on, multiple-input-multiple-output (MIMO) antenna technology is applied since it can provide a high data rate without the need for additional bandwidth [1]. Additionally, as the IoT devices are becoming smaller, while simultaneously demanding robust processing capabilities, there is a strong demand for a large-scale MIMO antenna with compact size. In this context, the microstrip patch antenna array is commonly used in many MIMO systems because of its low profile and ease of integration as well.

In general, the task of integrating numerous microstrip patches into compact IoT devices remains highly challenging. The most important part of achieving a compact-size MIMO antenna is a decoupling structure, which is employed for the surface-wave or space-wave coupling suppression in the patch array. In [2–7], the decoupling structures directly above the

**Data Availability Statement:** All relevant data are within the paper.

**Funding:** The Vietnam Academy of Science and Technology (VAST) under grant number TANQP.02/23-25. The funders had no role in study

design, data collection and analysis, decision to publish, or preparation of the manuscript.

**Competing interests:** The authors have declared that no competing interests exist.

antenna layer could increase isolation by suppressing the space-wave coupling. They could be a near-field resonator (NFR) [2, 3], an array antenna decoupling surface [4], a metasurface (MS) [5, 6], an asymmetrical coplanar strip wall [7] and a dielectric block [8]. Although high isolation with small element spacing can be achieved, these decoupling types always require a large distance from the antenna layer. Accordingly, the antennas suffer from a very high profile of higher than $0.15\lambda$, which makes them not practically appealing for low-profile IoT devices.

To tackle the abovementioned technical hurdle, the defected ground structure (DGS) performing as a band-stop filter is used to suppress the surface-wave current. It is introduced on the ground plane for isolation enhancement [9–11]. Although low-profile configuration can be achieved, the ground modification causes negative effects on the antenna radiation, such as decreasing the realized peak gain and increasing the backward radiation. Alternatively, electromagnetic bandgap (EBG) [12–14], split ring resonator (SRR) [15, 16], mixed radiation modes [17], and self-decoupling [18] are also effective methods for surface-wave suppression. In [19], the mutual coupling can be suppressed by exciting orthogonal mode on the non-excited element. These decoupling structures are occupied between the MIMO elements, leading to a large element spacing greater than $0.03\lambda$. In [20, 21], smaller element spacing is obtained with the help of a grounded stub or a lumped inductance.

In summary, even if the aforementioned MIMO antennas can attain good isolation performance, their applications are probably limited by a few inherent drawbacks, such as high profile and large element spacing. Most are also two-element MIMO or large-scale 1-D MIMO array systems. The motivation of this paper is to design a large-scale 2-D MIMO antenna with a low profile and small element spacing characteristics. For a better understanding of the design procedure, the decoupling mechanism is first discussed. Then, the configurations and decoupling networks for the H-plane and E-plane MIMO arrays are presented. Finally, the measurements are implemented to verify the proposed concept. The antenna is simulated using the commercial High-Frequency Structure Simulator (HFSS).

## Decoupling mechanism

The excited and non-excited patches of an arbitrary MIMO antenna can be considered as transmitting and receiving elements for a MIMO antenna. Assume that the electric field from the excited patch is the incoming wave with polarization $\mathbf{E}_i$. The polarization of the electric field of the receiving antenna (non-excited patch) is $\mathbf{E}_r$.

$$\mathbf{E_i} = \widehat{\rho_w} E_i \tag{1}$$

$$\mathbf{E_r} = \widehat{\rho_r} E_r \tag{2}$$

where $\widehat{\rho_w}$ and $\widehat{\rho_r}$ are the unit vectors of the incoming wave and the polarization vector, respectively, as shown in Fig 1. The polarization loss factor (PLF) characterizes the loss of electromagnetic power due to the polarization mismatch [22].

$$\boldsymbol{PLF} = |\widehat{\rho_w} \cdot \widehat{\rho_r}|^2 = |\cos\varphi|^2 \tag{3}$$

In this scenario, the mutual coupling is the power captured by the receiving patch from the transmitting patch. This can be expressed through the PLF, as presented in Fig 1. When transmitting and receiving antennas are of different polarizations, for example, with vertical and horizontal polarizations, respectively. In this case, PLF is equal to 0 and no power is transferred between the antennas. In contrast, when the polarizations are similar, the PLF is equal to 1 and the receiving antenna is possible to capture maximum power from the incident wave.

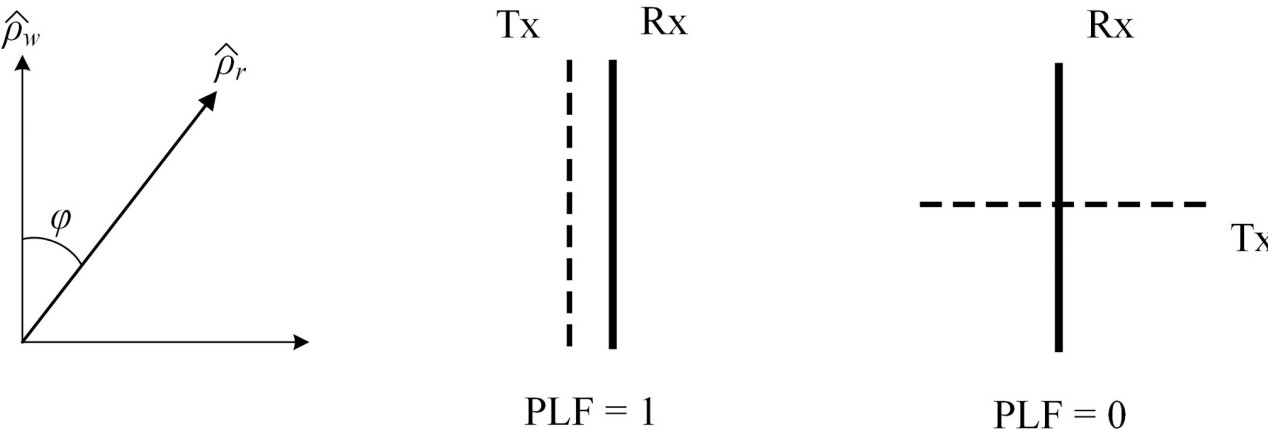

**Fig 1. The PLF of different transmitting and receiving polarizations.**

In the MIMO antenna, when two patches are positioned in close proximity, the polarization of the non-excited element will be similar to that of the excited element. Thus, high mutual coupling is observed. To suppress this mutual coupling, an effective wave is to change the coupling path between the MIMO elements so that they are working in orthogonal modes. This method is used to decouple the H-plane and E-plane MIMO arrays, which are presented in the following Sections.

## Two-element H-plane MIMO array

### Antenna design

The geometry of a two-element MIMO antenna is shown in Fig 2. Two microstrip patches are arranged in the H-plane configuration. The coaxial cables are used to feed the antenna. The used dielectric substrate is Taconic TLY-5 with relative permittivity of 2.2 and a loss tangent of 0.0009. To decouple the antenna, a microstrip line is added in close proximity to the radiating edges of the patches. The specific dimensions of the H-plane MIMO antenna are as follows: $L = 80$, $W = 50$, $H = 0.76$, $L_p = 20$, $W_p = 23.6$, $l_f = 4$, $d_h = 1.6$, $s = 1.0$, $l = 22.5$, $w = 0.9$ (unit: mm).

The performance comparison in terms of the reflection coefficient $|S_{11}|$ and the transmission coefficient $|S_{21}|$ of the coupled and decoupled MIMO antennas is shown in Fig 3. These designs are optimized for a similar operating frequency at 4.8 GHz. As seen, the isolation of the coupled MIMO is very high around 8 dB. With the presence of the decoupling structure, the isolation is significantly improved to 32 dB.

### Antenna characteristics

According to the abovementioned theory, it can be concluded that when two MIMO antennas are in close proximity, the mutual coupling can be suppressed if they operate in orthogonal modes. The simulated current distributions at 4.8 GHz for both coupled and decoupled MIMO antennas are illustrated in Fig 4. For the coupled MIMO antenna, the coupling occurs by the fringing fields to the nearby edges. The induced current on the non-excited patch is in the same direction as the current flowing on the excited patch. In this case, these patches have similar polarization and the PLF is equal to 1. Thus, the mutual coupling for the coupled MIMO antenna is very high. In order to suppress the mutual coupling, the PLF should be 0. This condition can be satisfied by using a microstrip line positioned along the radiating edges

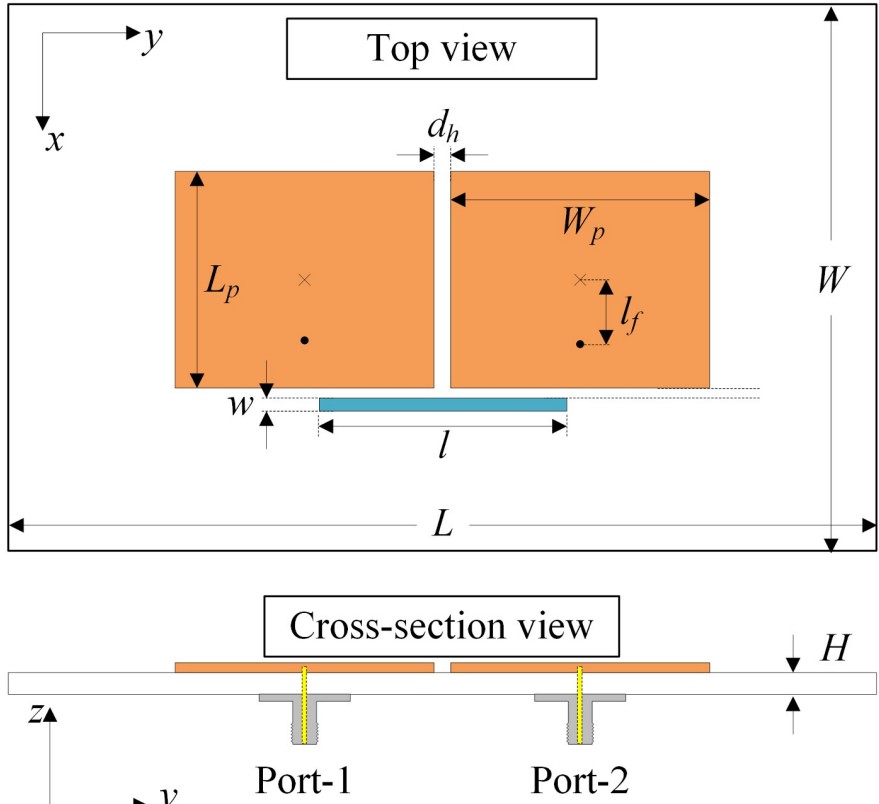

**Fig 2. Geometry of the H-plane MIMO array.**

of the patches. There is a coupling signal from the excited patch to the microstrip line and to the non-excited patch. Therefore, the current flowing on the non-excited patch is orthogonal with that on the excited element. This contributes to suppressing the mutual coupling between the MIMO elements.

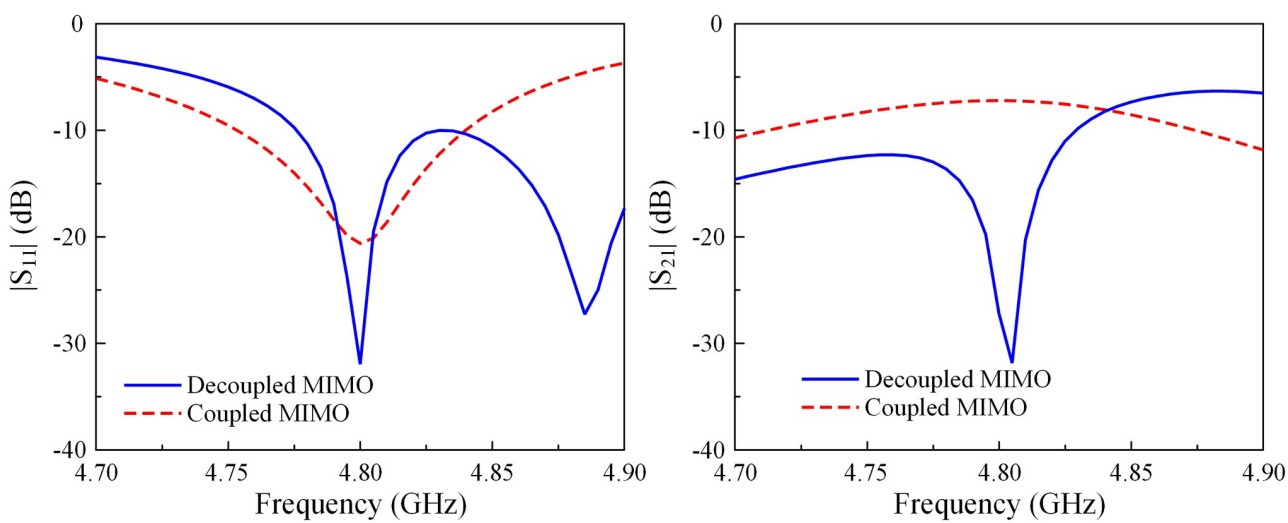

**Fig 3. Simulated $|S_{11}|$ and $|S_{21}|$ of the coupled and decoupled H-plane MIMO array.**

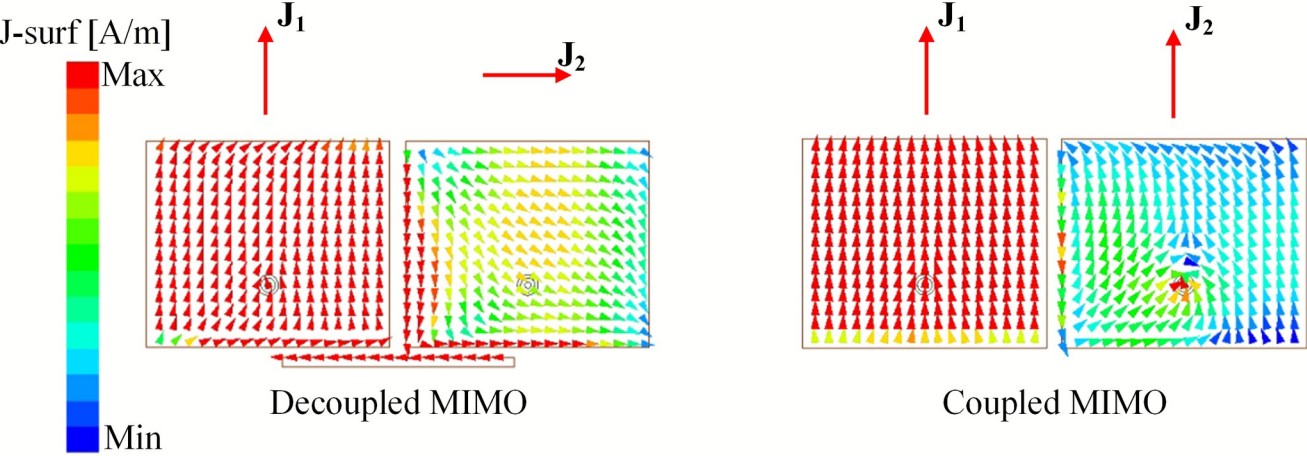

**Fig 4. Simulated current distributions of the coupled and decoupled H-plane MIMO array.**

In order to optimize this antenna, several key parameters are related to the position ($s$) and the length ($l$) of the decoupling structure. Figs 5 and 6 show the simulated reflection and transmission coefficients of the proposed two-element H-plane MIMO array against the variations of $s$ and $l$. Since the antenna works as an LC resonant circuit, decreasing $s$ will increase the capacitance formed by the patches and the microstrip line. Meanwhile, increasing $l$ will both increase the capacitance and the inductance values. Thus, the $|S_{11}|$ resonances in move towards the lower operating frequency band. Similar behaviors are also observed for isolation performance.

## Two-element E-plane MIMO array

Since the E-field distribution on radiating edge of the microstrip patch antenna is different from that on the non-radiating edge, the coupling scenario in the E-plane should be different

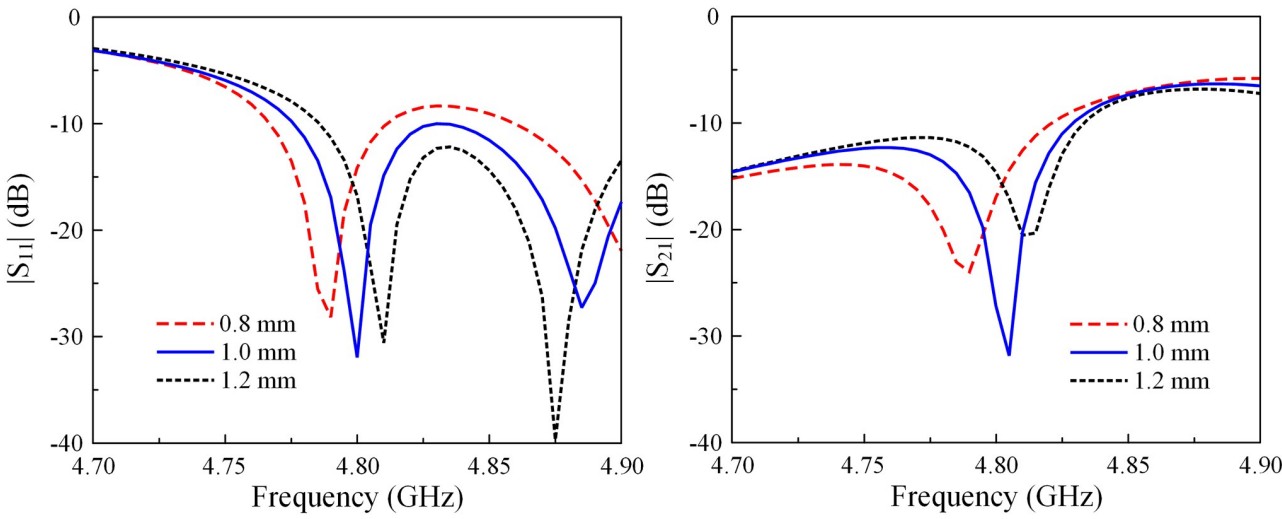

**Fig 5. Simulated $|S_{11}|$ and $|S_{21}|$ of the H-plane MIMO array for different values of $s$.**

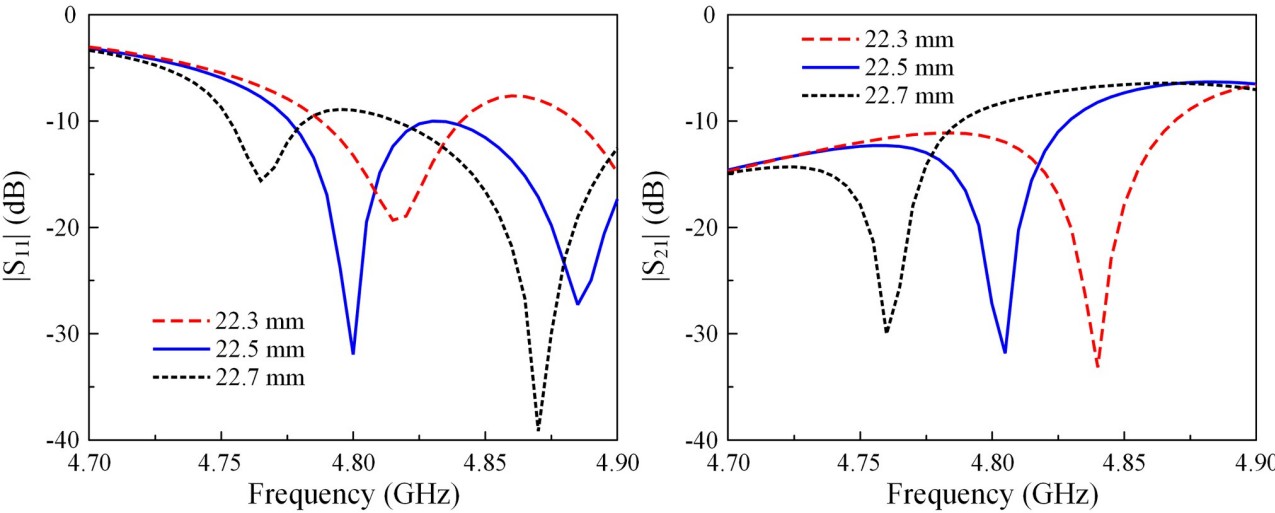

**Fig 6. Simulated $|S_{11}|$ and $|S_{21}|$ of the H-plane MIMO array for different values of $l$.**

from the H-plane. Therefore, the decoupling mechanism is varied and in this section, an E-plane coupled two-element MIMO array is investigated.

## Antenna design

The geometry configuration in terms of the top view and the side view of a two-element MIMO array is presented in Fig 7. The patches are arranged in the E-plane configuration with an edge-to-edge distance of 2.0 mm, corresponding to 0.032 λ at 4.8 GHz. The antenna is also fed by two coaxial cables. A grounded microstrip line is positioned between the MIMO elements for mutual coupling suppression. The optimized dimensions of the two-element E-plane MIMO array are given as follows: $L = 80$, $W = 50$, $H = 0.76$, $L_p = 20$, $W_p = 20.4$, $l_f = 2.8$, $d_e = 2.0$, $r_v = 0.3$, $l_0 = 11.0$, $w_0 = 1.0$ (unit: mm).

A performance comparison with respect to reflection and transmission coefficients for different MIMO antennas is depicted in Fig 8. Note that the decoupling structure is not utilized in the coupled MIMO design. This antenna has a resonance at 4.8 GHz with very low isolation of 8 dB. On the other hand, the decoupled MIMO array can achieve higher isolation of 19 dB with the aid of the decoupling structure.

## Antenna characteristics

According to the theory discussed in Fig 1, the operating modes on these patches are similar when two patches are arranged in the E-plane configuration. Thus, strong mutual coupling is observed. In order to suppress the mutual coupling, a quarter-wavelength shorted stub is employed. The function of this decoupling structure is to alter the operating mode on the non-excited element, which changes from a similar mode to the orthogonal mode with the excited element. This method has been proposed in [19], but the authors have just investigated the effect of the shorted quarter-wavelength on the H-plane MIMO array. Further demonstration of the effectiveness of this method on decoupling the E-plane MIMO array is presented in this paper.

The mutual coupling reduction can be demonstrated by investigating the antennas' current distribution with and without the decoupling network. Fig 9 shows the current distributions at

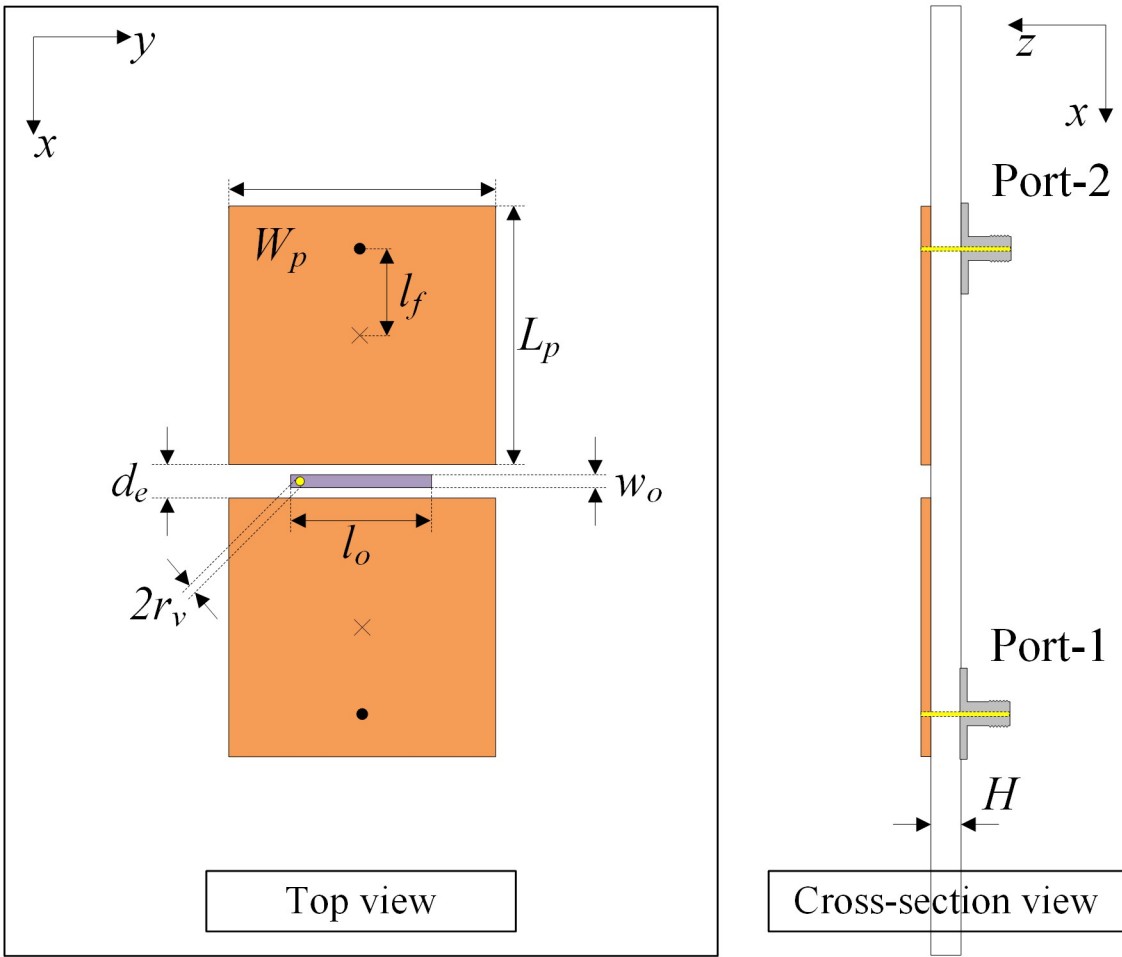

**Fig 7. Geometry of the proposed E-plane MIMO antenna.**

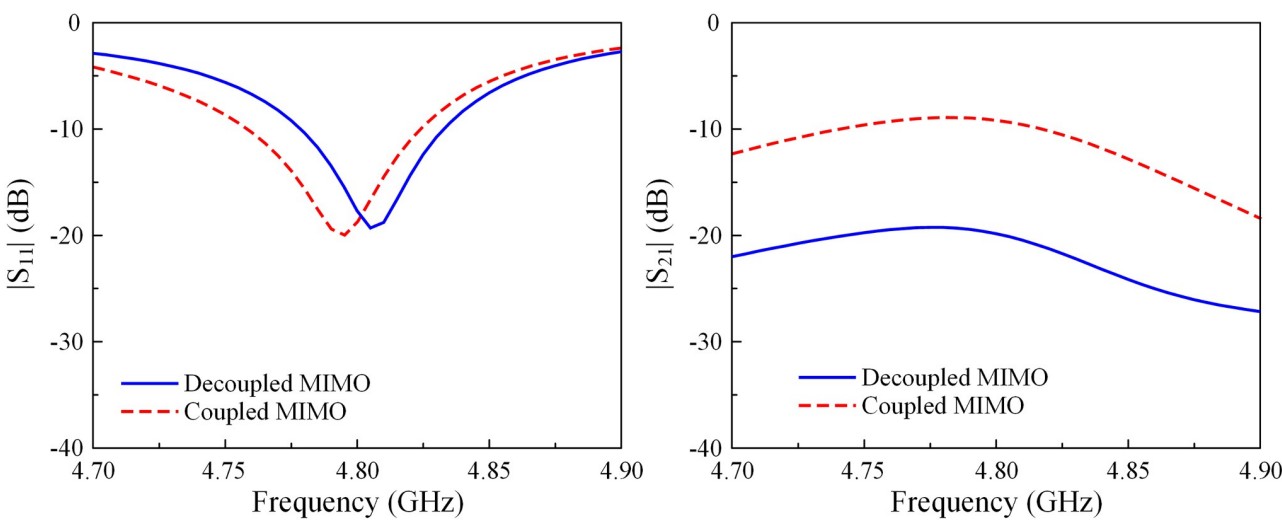

**Fig 8. Simulated $|S_{11}|$ and $|S_{21}|$ of the coupled and decoupled E-plane MIMO array.**

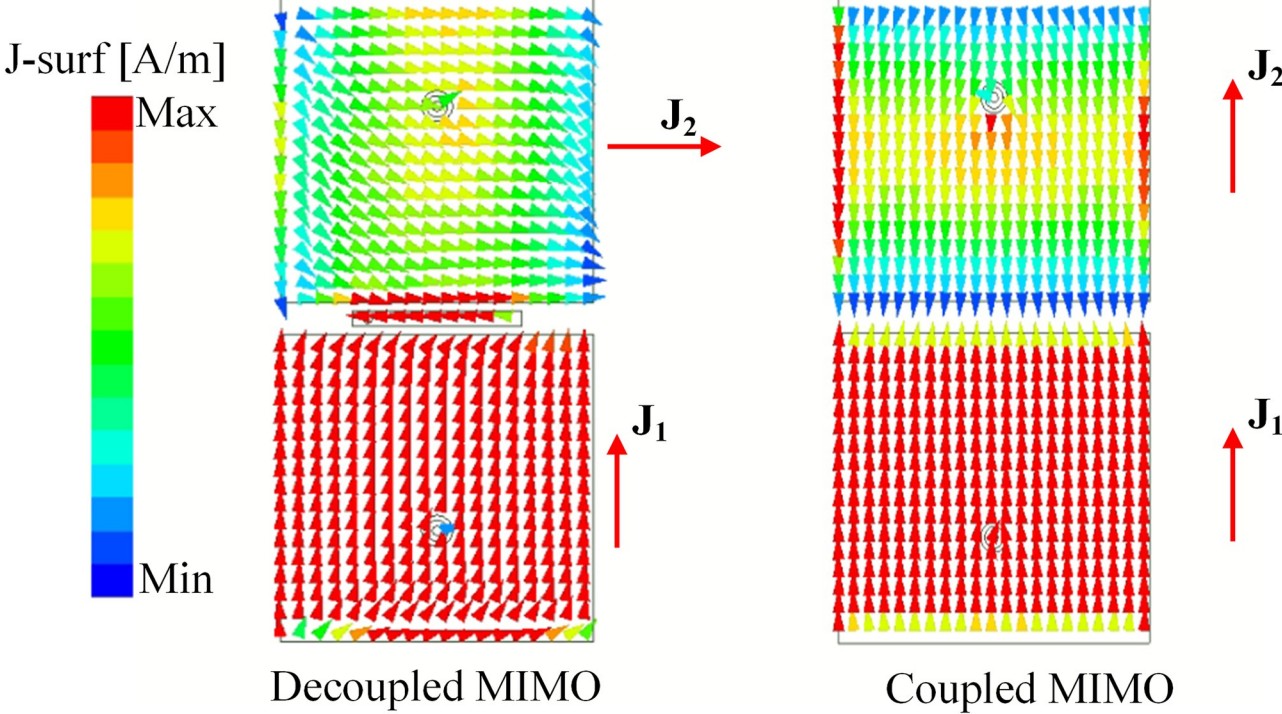

**Fig 9. Simulated current distributions of the coupled and decoupled E-plane MIMO array.**

4.8 GHz of the coupled and decoupled MIMO antennas. As observed, a current flowing on the non-excited element is similar to that on the excited one, leading to high isolation. On the other hand, the presence of the grounded stub significantly suppresses the mutual coupling between the MIMO elements. In this case, the current on the non-excited element is orthogonal to that on the excited element. Thus, the mutual coupling is mitigated.

Several key parameters are related to the radius of the vias ($r_v$) and the length of the stub ($l_0$) decoupling structure to optimize the E-plane MIMO antenna. As shown in Figs 10 and 11, the effects of these parameters on the matching performance are insignificant. On the other hand, isolation suffers from the significant impact of these parameters.

## Extension to a large-scale array

### 2 × 2 MIMO array

In this section, the proposed decoupling methods are further extended to a large-scale MIMO array. Fig 12 shows a 2 × 2 planar MIMO patch array. The decoupling structures to suppress the mutual coupling in both E- and H-plane are presented in the previous sections. Additionally, the edge-to-edge distance $d_e$ and $d_h$ are fixed as optimal values of two-element H- and E-plane MIMO arrays. The optimized dimensions of the 2 × 2 planar MIMO patch array are as follows: $L = 80$, $W = 70$, $L_p = 19.8$, $W_p = 23.6$, $l_f = 4.2$, $d_e = 2.0$, $d_h = 1.6$, $r_v = 0.3$, $l_0 = 11.0$, $w_0 = 1.0$, $s = 1.0$, $l = 22.5$, $w = 0.9$ (unit: mm).

Fig 13 shows the simulated S-parameter of the proposed 2 × 2 planar MIMO patch array. It can be seen that around 4.8 GHz, the antenna shows good matching and isolation performances. The isolations among the MIMO ports are less than 20 dB at 4.8 GHz. The simulated radiation patterns at 4.8 GHz are also illustrated in Fig 13. Note that the radiation patterns are

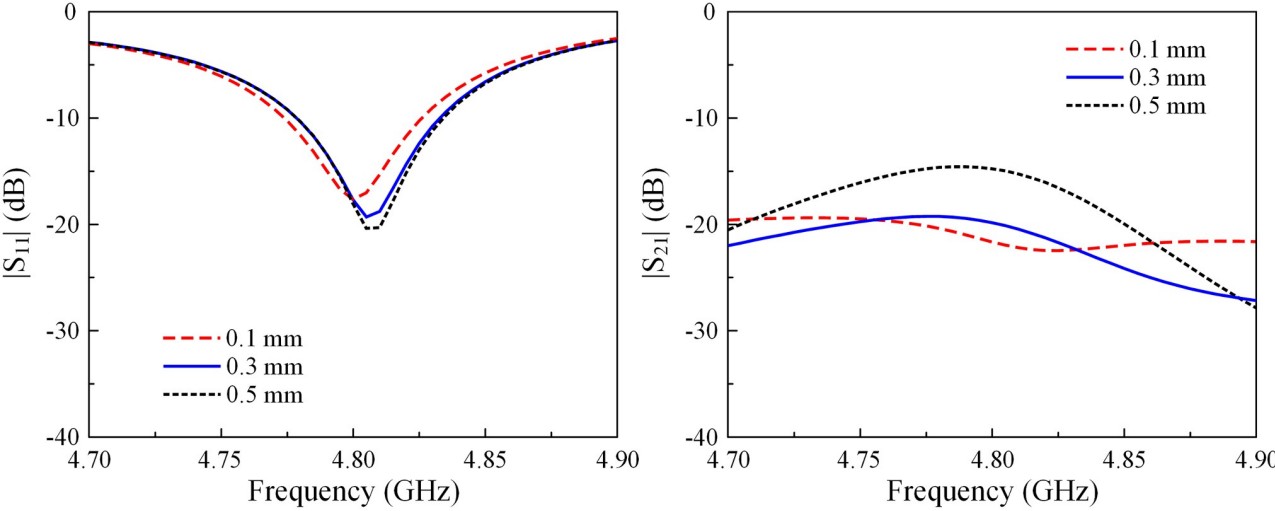

**Fig 10. Simulated $|S_{11}|$ and $|S_{21}|$ of the E-plane MIMO array for different values of $r_v$.**

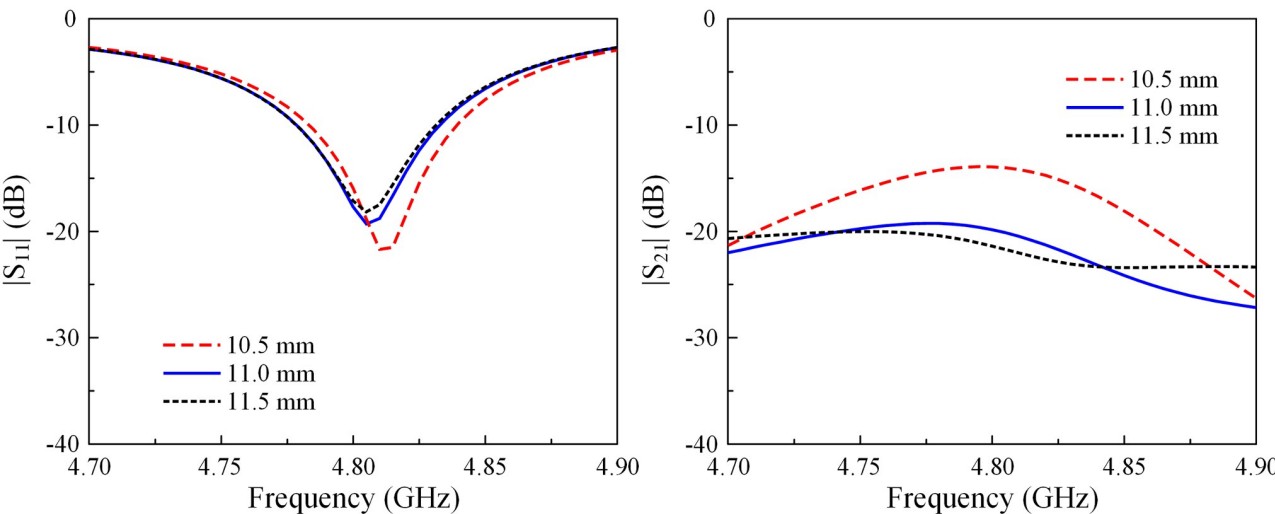

**Fig 11. Simulated $|S_{11}|$ and $|S_{21}|$ of the E-plane MIMO array for different values of $l_0$.**

similar for all ports due to the symmetrical geometry. The E- and H-plane radiation patterns demonstrate that the proposed $2 \times 2$ MIMO array has good radiation characteristics. The radiation is quite symmetric around the broadside direction with the broadside gain of better than 4.5 dBi. Additionally, the cross-polarization in this direction is also significantly less than the co-polarization, which is −14.3 dB in comparison with 4.6 dB.

## $2 \times N$ MIMO array

In this phase, the feasibility of the proposed decoupling configurations in a large-scale MIMO array is further examined by assessing their potential for extension in the H-plane. For this purpose, a larger $2 \times N$ MIMO array is studied. The operation of the $2 \times N$ MIMO array will

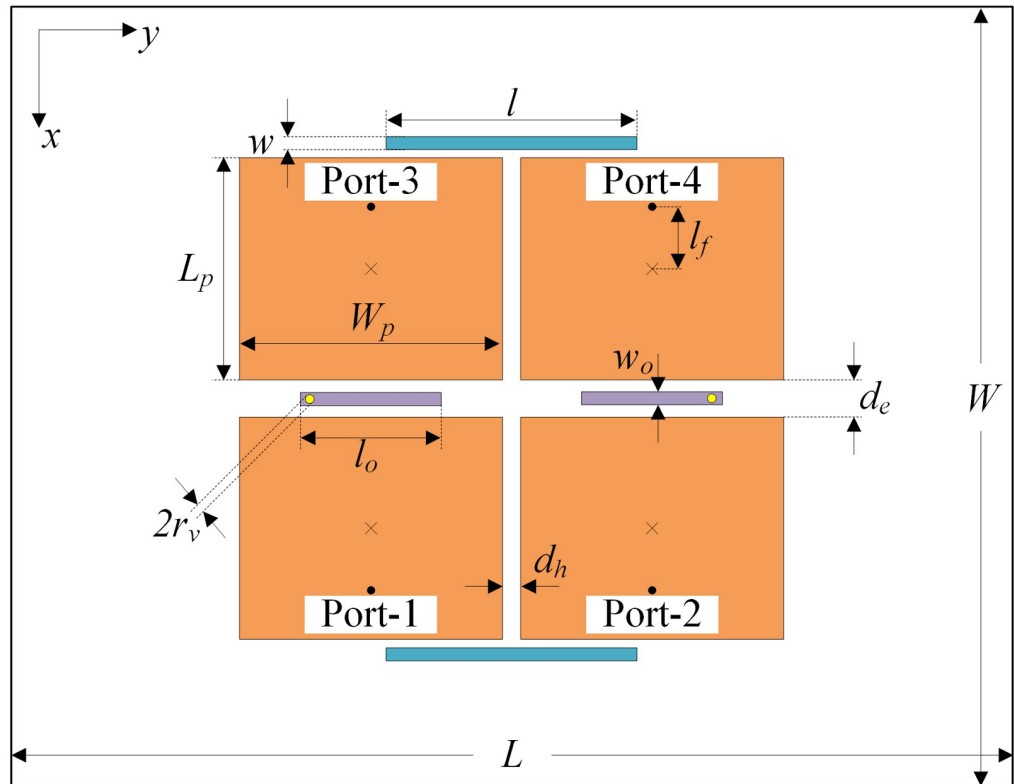

**Fig 12. Geometry of the proposed 2 × 2 MIMO antenna array.**

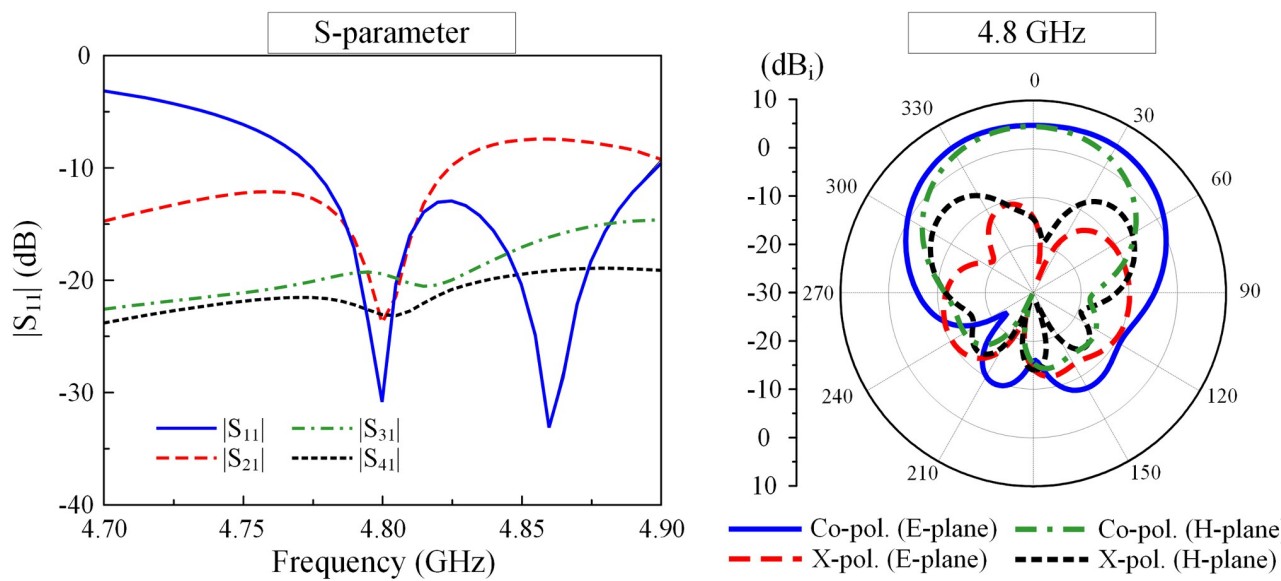

**Fig 13. Simulated S-parameters and radiation patterns at 4.8 GHz of the proposed 2 × 2 MIMO array with Port-1 excitation.**

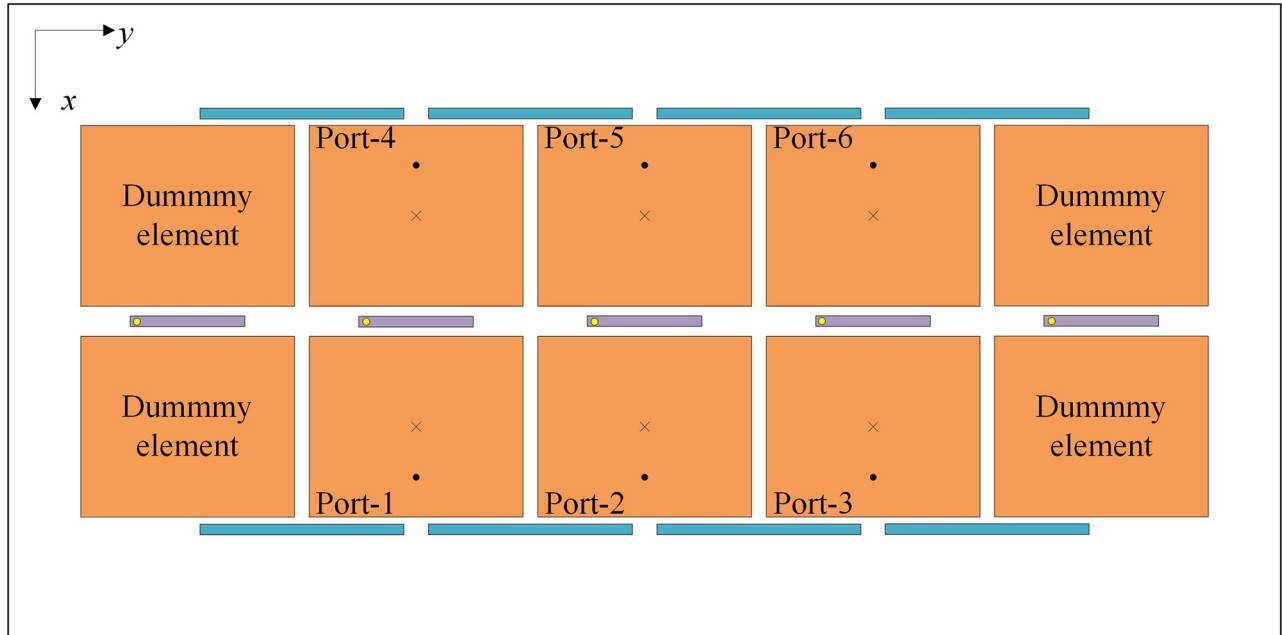

**Fig 14. Geometry of the proposed 2 × 3 MIMO antenna array.**

be similar to that of the 2 × 3 MIMO array as shown in Fig 14. For this structure, four dummy elements are placed at both sides of the array to achieve identical results for all ports. Notably, in this design, only the substrate is enlarged to $150 \times 70 \ mm^2$ while other design parameters remain unchanged.

The simulated S-parameters and radiation patterns of the proposed 2 × 3 MIMO are shown in Fig 15. The results for other ports are similar due to the symmetry of the array structure, and therefore they are not shown for brevity. It can be observed that both Port-1 and Port-2 are well matched at 4.8 GHz. The overlap between −10 dB impedance and isolation bandwidths is from 4.78 to 4.81 GHz, which is quite similar to the 2 × 2 MIMO antenna (4.78–4.83 GHz). Across this band, the coupling coefficient between any two elements is below −20 dB, demonstrating a good decoupling effect. In addition, the radiation patterns at 4.8 GHz for Port-1 and Port-2 excitations shown in Fig 15 also demonstrate the good radiation characteristics of the 2 × 3 MIMO array. Accordingly, the antenna will definitely work with an extensive increase in the number of elements in the H-plane to form a large-scale 2 × N MIMO array.

## Measurement results and discussion

In this Section, the 2 × 2 MIMO array is fabricated and measured to demonstrate the feasibility of the utilized decoupling structures. The photographs in terms of top-view and bottom-view of the fabricated antenna prototype are presented in Fig 16. An Agilent E8362B network analyzer is employed to measure the S-parameters in an open-air environment. Far-field radiation assessments are conducted within an anechoic chamber situated at the Electromagnetic Wave Technology Institute in Seoul, Korea. The proposed antenna serves as the receiving antenna, while a standard wideband horn antenna functions as the transmitting antenna. In general, the simulations and measurements exhibit a close correspondence. The small difference might be caused by the tolerances in fabrication and imperfections in the measurement setup.

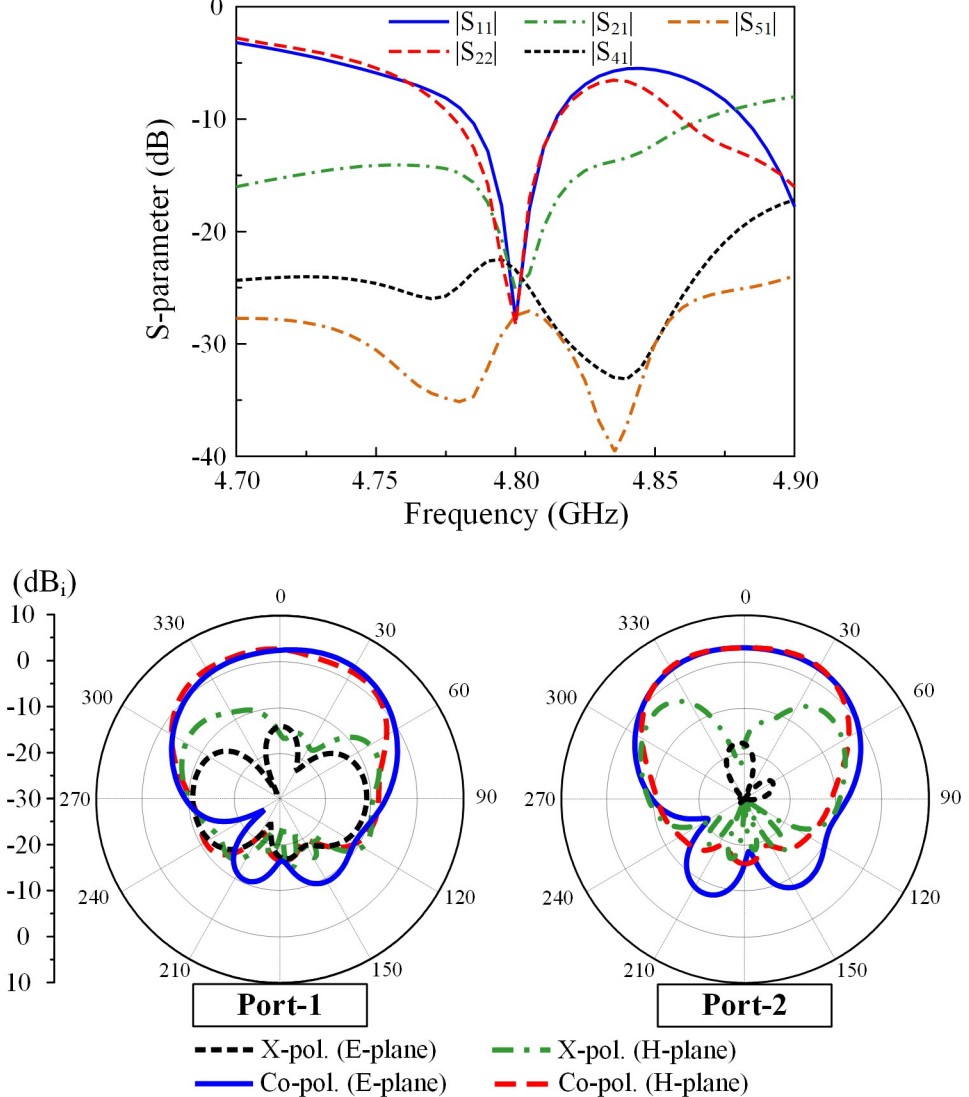

**Fig 15. Simulated S-parameters and radiation patterns of the proposed 2 × 3 MIMO antenna array.**

### S-parameter and far-field results

Fig 17 illustrates the comparison between the simulated and measured S-parameters of the fabricated MIMO array. The proposed antenna meets the criteria for −10 dB impedance matching within the frequency range of 4.76 to 4.83 GHz. At 4.8 GHz, the inter-port isolations are all lower than −20 dB. Besides, the operating frequency range with isolation of better than 15 dB and reflection coefficient of less than −10 dB is from 4.78 to 4.81 GHz.

Due to the symmetrical antenna geometry, the far-field performances are only characterized by Port-1 excitation. Fig 18 presents the realized gain results in the broadside direction (+z-direction). For the far-field test, when one port is excited, the other is terminated with a 50-Ω load. As observed, the simulated and measured gain values are almost similar. The antenna has a maximum gain of 4.5 dBi within the operating band from 4.78 to 4.81 GHz. The radiation patterns at 4.8 GHz of the proposed antenna are plotted in Fig 19. It can be seen that

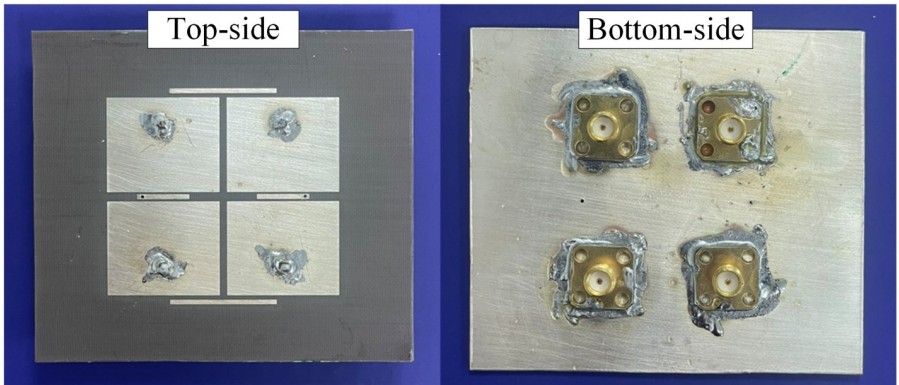

**Fig 16. Photographs of the fabricated proposed 2 × 2 MIMO antenna array.**

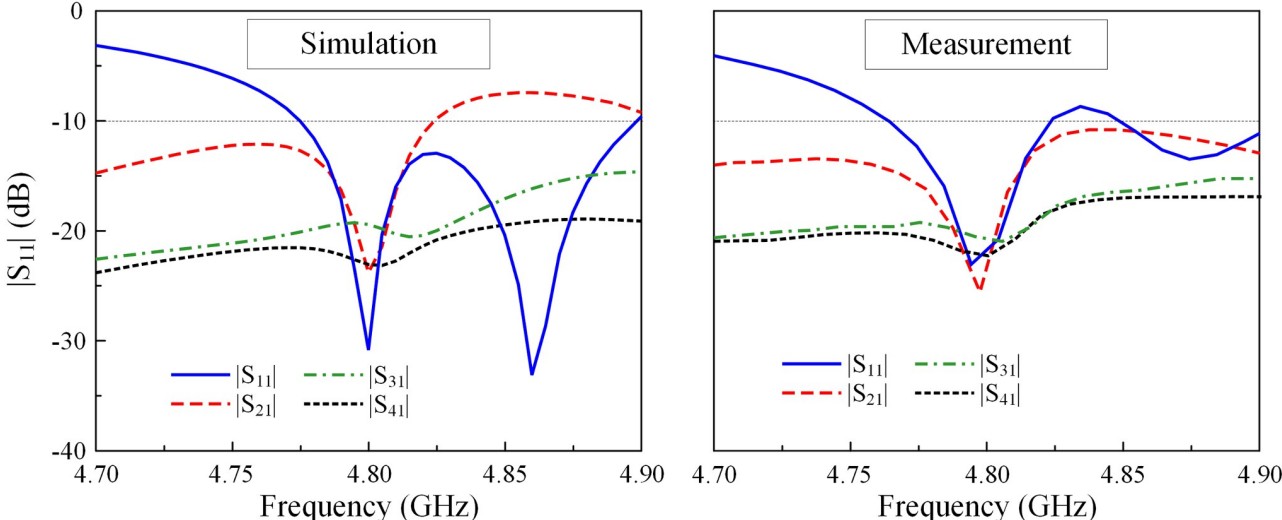

**Fig 17. Measured S-parameters of the proposed 2 × 2 MIMO antenna array.**

in both E- and H-plane, the proposed antenna performs good bore-sight radiation. The cross-polarization is about 16 dB less than the co-polarization and the front-to-back ratio is higher than 15 dB.

## MIMO performances

The envelope correlation coefficient (ECC), the diversity gain (DG), the mean effective gain (MEG), the channel capacity loss (CCL) are the most important MIMO parameters. They are calculated based on the S-parameter and far-field as Eqs (4)–(7) [23].

$$ECC_{ij} = \frac{|R_{ii}^* * T_{ij} + T_{ji}^* * S_{jj}|^2}{(1 - |R_{ii}|^2 - |T_{ji}|^2)(1 - |R_{jj}|^2 - |T_{ij}|^2)} \qquad (4)$$

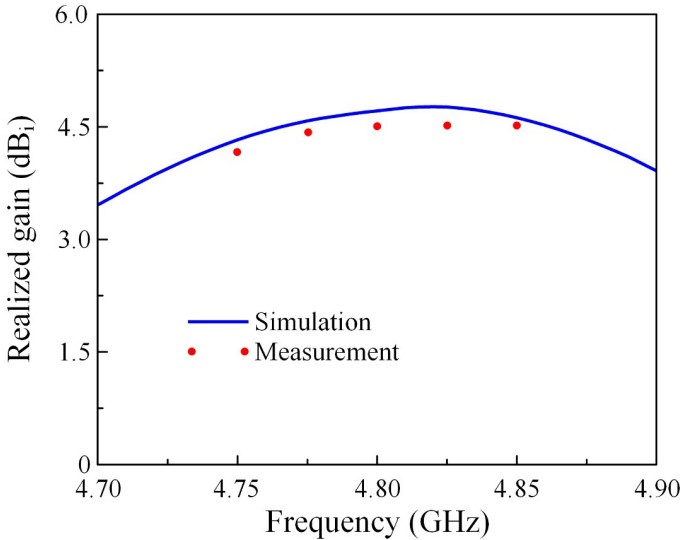

**Fig 18. Measured realized gain of the proposed 2 × 2 MIMO antenna array.**

$$D_{gain} = 10\sqrt{1 - |ECC_{ij}|^2} \tag{5}$$

$$MEG_i = 0.5\left(1 - \sum_{j=1}^{K} |R_{ij}|\right) \tag{6}$$

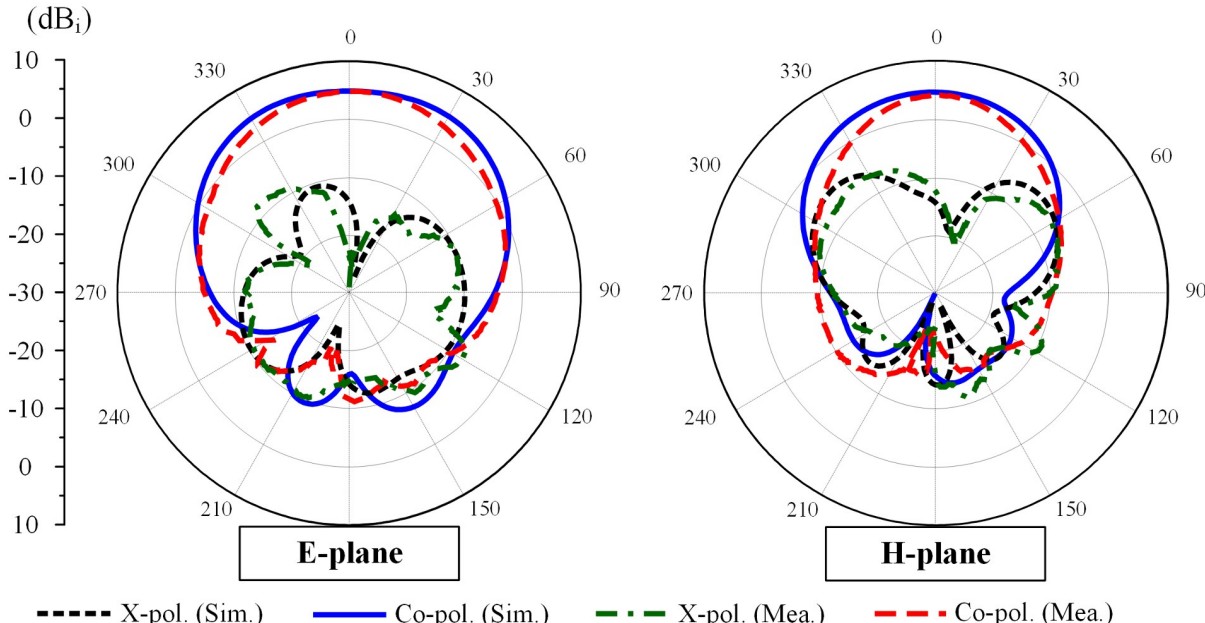

**Fig 19. Measured radiation patterns at 4.8 GHz of the proposed 2 × 2 MIMO antenna array.**

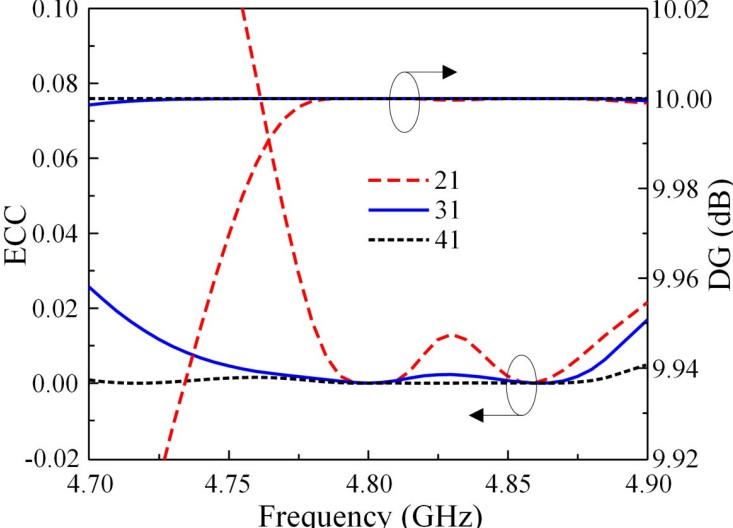

**Fig 20. Calculated ECC and DG of the proposed 2 × 2 MIMO antenna array.**

$$CCL = -2 \log_2 \det\left(\begin{bmatrix} \rho_{ii}\rho_{ij} \\ \rho_{ji}\rho_{jj} \end{bmatrix}\right); \qquad \begin{aligned} \rho_{ii} &= 1 - \left(|R_{11}|^2 - R_{ij}{}^2\right) \\ \rho_{ij} &= -\left(R_{ii} * R_{ij} - R_{ji} * R_{jj}\right) \end{aligned} \tag{7}$$

Here, $i$ and $j$ are port numbers. $R$ and $T$ are the reflection and transmission coefficients. The calculated ECC and DG results are illustrated in Fig 20. It can be seen that the observed ECC is significantly below the allowable threshold of 0.5 throughout the operating frequency range. The DG is nearly at its theoretical maximum value of approximately 10. Regarding the MEG and CCL, the data in Fig 21 indicates that the MEG is less than 0.5 and the CCL is smaller than 0.4 bps/Hz.

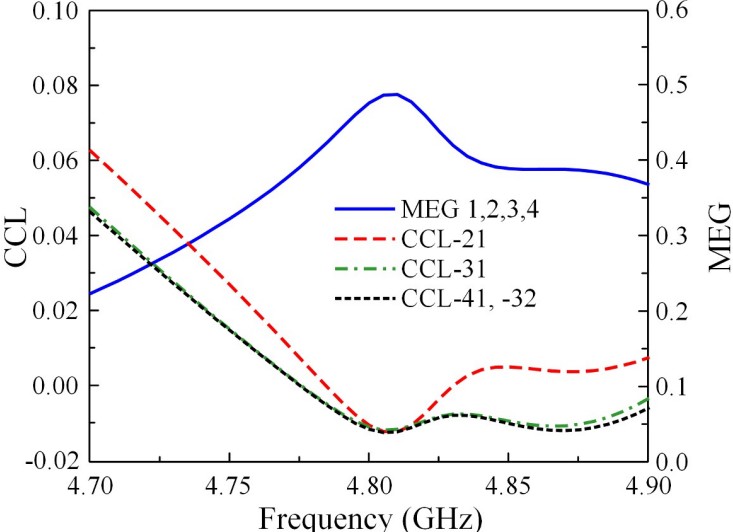

**Fig 21. Calculated MEG and CCL of the proposed 2 × 2 MIMO antenna array.**

**Table 1. Performance comparison among microstrip patch MIMO array antennas.**

| Ref. | Resonance (GHz) | Profile ($\lambda$) | Array config. | Extended array | Edge-to-edge Spacing ($\lambda_c$) | Max. Isolation (dB) |
|---|---|---|---|---|---|---|
| [2] | 3.7 | 0.22 | 1 × 2 | No | 0.034 | 33 |
| [5] | 2.6 | 0.05 | 1× 2 | No | 0.008 | 38 |
| [7] | 5.8 | 0.22 | 1 × 2 | No | 0.03 | 54 |
| [8] | 5 | 0.18 | 1 × 2 | No | 0.100 (E-plane) | 50 |
|  | 10 |  |  |  | 0.027 (H-plane) | 25 |
| [10] | 5 | 0.28 | 1 × 2 | No | 0.033 | 34 |
| [15] | 5.2 | 0.02 | 1 × 2 | No | 0.039 | 56 |
| [17] | 2.5 | 0.05 | 2 × 2 | No | 0.061 (E-plane) | 27 |
|  |  |  |  |  | 0.061 (H-plane) |  |
| [18] | 3.5 | 0.04 | 1 × 2 | 1 × N | 0.183 | 55 |
| [19] | 5.8 | 0.03 | 1 × 2 | 1 × N | 0.07 | 24 |
| [20] | 3.2 | 0.02 | 1 × 2 | 1 × N | 0.027 | 18 |
| [21] | 2.5 | 0.1 | 1 × 2 | N × N | 0.016 | 15 |
| Prop. | 4.8 | 0.01 | 2× 2 | 2× N | 0.032 (E-plane) | 25 |
|  |  |  |  |  | 0.026 (H-plane) |  |

## Comparison with related works

To demonstrate the effectiveness of the proposed decoupling structures, Table 1 summarizes the comparison among microstrip patch MIMO antennas. In [2, 5, 7, 8, 10, 15], the proposed decoupling techniques are useful in decoupling 2-element MIMO. In [17], better extension to 2 × 2 MIMO array is achieved. It can be seen that there are several designs, which can be extended to large-scale MIMO array [18–21]. In general, competitive isolation performance in both E- and H-plane can be realized by the proposed antenna. Although high isolation can be achieved in [18, 19], these designs suffer from a critical drawback of extremely large spacing. Meanwhile, the MIMO arrays in [20, 21] have smaller spacing, but low isolation. Additionally, low profile is also another attractive feature of the proposed antenna.

## Conclusion

This paper introduces a low-profile and large-scale MIMO antenna system. The antenna consists of multiple elements arranged in both E-plane and H-plane configurations. The inherent strong coupling between the H-plane elements is effectively mitigated using a half-wavelength microstrip line, while the coupling in the E-plane is suppressed with the help of a grounded stub. The proposed 2 × 2 MIMO array exhibits an impedance BW spanning from 4.78 to 4.81 GHz and the isolation between the elements is better than 15 dB. This impressive performance is achieved with extremely small edge-to-edge distances of 0.032$\lambda$ and 0.026$\lambda$ in the E-plane and H-plane, respectively. Notably, this antenna design can be easily extended to a large-scale 2 × N MIMO array. In comparison with the other designs, the proposed MIMO array possesses a simple decoupling structure, small element spacing, and the capability of extending to large-scale arrays.

## Author Contributions

**Conceptualization:** Hung Tran-Huy, Thuy Nguyen Thi.

**Investigation:** Hung Tran-Huy, Thuy Nguyen Thi, Muhammad Aslam, Tung The-Lam Nguyen.

**Methodology:** Hung Tran-Huy, Tung The-Lam Nguyen.

**Supervision:** Hong Nguyen Tuan.

**Validation:** Muhammad Aslam, Tung The-Lam Nguyen.

**Writing – original draft:** Muhammad Aslam, Tung The-Lam Nguyen.

**Writing – review & editing:** Hung Tran-Huy, Hong Nguyen Tuan.

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
