## [Decision Letter · Decision Letter 0]

11 Sep 2023

PONE-D-23-25130A large-scale MIMO antenna system for 5G IoT applicationsPLOS ONE

Dear Dr. Tran-Huy,

Thank you for submitting your manuscript to PLOS ONE. After careful consideration, we feel that it has merit but does not fully meet PLOS ONE’s publication criteria as it currently stands. Therefore, we invite you to submit a revised version of the manuscript that addresses the points raised during the review process.

We look forward to receiving your revised manuscript.

Kind regards,

Musa Hussain

Academic Editor

PLOS ONE

Journal Requirements:

Additional Editor Comments:

Improve the English.

Reviewers' comments:

Reviewer's Responses to Questions

**Comments to the Author**

1. Is the manuscript technically sound, and do the data support the conclusions?

Reviewer #1: Yes

Reviewer #2: Yes

2. Has the statistical analysis been performed appropriately and rigorously? 

Reviewer #1: Yes

Reviewer #2: Yes

3. Have the authors made all data underlying the findings in their manuscript fully available?

Reviewer #1: Yes

Reviewer #2: Yes

4. Is the manuscript presented in an intelligible fashion and written in standard English?

Reviewer #1: Yes

Reviewer #2: Yes

5. Review Comments to the Author

Reviewer #1: The authors presented a closely spaced MIMO antenna and its mutual coupling reduction with the help of a half-wavelength microstrip line and a quarter-wavelength grounded stub. The manuscript is well-written and the data has been organized well. The presented results are quite interesting. However, the reviewer has some comments for the improvements.

(1) What is the log tangent value of the antenna substrate? Did authors use the same substrate and thickness for the all MIMO designs?

(2) Comment on the measurement setup of the antenna.

(3) Cite a proper reference for the equations used for the computation of the ECC/DG etc.

(4) What is the unit of antenna profile in Table 1? Is it mm or the wavelength ?

(5) The maximum isolation of the MIMO antenna is more than 25 dB as it is evident from the Figure 17. Why did the authors have stated maximum isolation as 20 dB in Table 1? Please explain.

(6) Which shortwave was used for the simulation?

Reviewer #2: The authors presented a low coupling MIMO antenna that can be easily extendable to a large-scale MIMO antenna system targeting 5G applications and IoT systems. The paper is well written, all the idea and concept is presented in professional way backed by strong mathematical explanation along with state-of-the-art.

There are few things that need to explain further:

1: Cite the reference from where the equations 1-3 are extracted.

2: Please comment on the bandwidth variation observed when the 2 x 2 MIMO array is extended to 2 X 3 MIMO array.

3: The decoupling is improved only at the desired bandwidth, what could be possible solution to improve the decoupling at the nearby frequency. Please comment.

4: The authors should cite the proper reference from where the equation 4 and 5 are extracted.

5: Since the work is focused on MIMO antenna, authors should add other MIMO performance parameters including Mean effective gain as explained in Highly selective multiple-notched UWB-MIMO antenna with low correlation using an innovative parasitic decoupling structure, 2023. Or the Channel capacity loss as explined in Self-decoupled tri band MIMO antenna operating over ISM, WLAN and C-band for 5G applications, 2023.

6: Explain the reason of discrepancy among simulated and measured results of s-parameters. How it can be minimise?

6. PLOS authors have the option to publish the peer review history of their article (what does this mean?). If published, this will include your full peer review and any attached files.

Reviewer #1: No

Reviewer #2: **Yes: **Wahaj Abbas Awan

---

## [Author Response · Author response to Decision Letter 0]

15 Sep 2023

Original Manuscript ID: PONE-D-23-25130

Original Article Title: “A large-scale MIMO antenna system for 5G IoT applications”

To: Reviewer

Re: Response to reviewer

Dear Reviewer,

We appreciate you for your precious time in reviewing our paper and providing valuable comments. It was your valuable and insightful comments that led to possible improvements in the current version. The authors have carefully considered the comments and tried our best to address every one of them.

We are uploading our point-by-point response to the comments, an updated manuscript with red highlighting indicating changes, and a manuscript without track changes.

Best regards,

 

Reviewer 1: The authors presented a closely spaced MIMO antenna and its mutual coupling reduction with the help of a half-wavelength microstrip line and a quarter-wavelength grounded stub. The manuscript is well-written and the data has been organized well. The presented results are quite interesting. However, the reviewer has some comments for the improvements.

Concern # 1: What is the log tangent value of the antenna substrate? Did authors use the same substrate and thickness for the all MIMO designs?

Author response: In this paper, the Taconic TLY-5 substrate is used for all designs. The loss tangent of this substrate is 0.0009.

Author action: The loss tangent is included in Paragraph 1, Section 3 of the revised manuscript.

Concern # 2: Comment on the measurement setup of the antenna.

Author response: Agreed.

Author action: The measurement setup is added to Paragraph 1, Section 6 of the revised manuscript.

Concern # 3: Cite a proper reference for the equations used for the computation of the ECC/DG etc.

Author response: Agreed.

Author action: The new references are added to the revised manuscript as ref [23].

Concern # 4: What is the unit of antenna profile in Table 1? Is it mm or the wavelength?

Author response: The antenna’s profile in Table 1 is defined with respect to lambda.

Author action: The unit for the profile is updated in Table 1 of the revised manuscript.

Concern # 5: The maximum isolation of the MIMO antenna is more than 25 dB as it is evident from the Figure 17. Why did the authors have stated maximum isolation as 20 dB in Table 1? Please explain.

Author response: The authors would like to thank the Reviewer for pointing out our mistake. In fact, 20 dB is the isolation among all ports of the MIMO antenna at 4.8 GHz. The maximum isolation is 25 dB.

Author action: The maximum isolation is modified in Table 1 of the revised manuscript.

Concern # 6: Which shortwave was used for the simulation?

Author response: The Ansys High-Frequency Simulation Structure (HFSS) is used in this paper.

Author action: The simulation tool is mentioned in Paragraph 4, Section 1 of the revised manuscript.

Reviewer 2: The authors presented a low coupling MIMO antenna that can be easily extendable to a large-scale MIMO antenna system targeting 5G applications and IoT systems. The paper is well written, all the idea and concept is presented in professional way backed by strong mathematical explanation along with state-of-the-art.

There are few things that need to explain further:

Concern # 1: Cite the reference from where the equations 1-3 are extracted.

Author response: Agreed.

Author action: Ref [22] is added to the revised manuscript.

Concern # 2: Please comment on the bandwidth variation observed when the 2 x 2 MIMO array is extended to 2 X 3 MIMO array.

Author response: Agreed.

Author action: Further discussion on the bandwidth variation is added to Paragraph 2, Section 5.2 of the revised manuscript.

Concern # 3: The decoupling is improved only at the desired bandwidth, what could be possible solution to improve the decoupling at the nearby frequency. Please comment.

Author response: The authors would like to thank the Reviewer for the very constructive comment. In fact, achieving high isolation in a wide frequency range is a very challenging task. The authors are also working on that. The solution could be to produce additional resonances in the S21 profile.

Concern # 4: The authors should cite the proper reference from where the equation 4 and 5 are extracted.

Author response: Agreed.

Author action: Ref [23] is added to the revised manuscript.

Concern # 5: Since the work is focused on MIMO antenna, authors should add other MIMO performance parameters including Mean effective gain as explained in Highly selective multiple-notched UWB-MIMO antenna with low correlation using an innovative parasitic decoupling structure, 2023. Or the Channel capacity loss as explained in Self-decoupled tri band MIMO antenna operating over ISM, WLAN and C-band for 5G applications, 2023.

Author response: Agreed.

Author action: The MEG and CCL are added to the revised manuscript as Fig. 21.

Concern # 6: Explain the reason of discrepancy among simulated and measured results of s-parameters. How it can be minimise?

Author response: Agreed.

Author action: Further discussion is added to Paragraph 1, Section 6 of the revised manuscript.

---

## [Decision Letter · Decision Letter 1]

19 Sep 2023

A large-scale MIMO antenna system for 5G IoT applications

PONE-D-23-25130R1

Dear Dr. Tran-Huy,

We’re pleased to inform you that your manuscript has been judged scientifically suitable for publication and will be formally accepted for publication once it meets all outstanding technical requirements.

Kind regards,

Musa Hussain

Academic Editor

PLOS ONE

Additional Editor Comments (optional):

Reviewers' comments:

Reviewer's Responses to Questions

**Comments to the Author**

1. If the authors have adequately addressed your comments raised in a previous round of review and you feel that this manuscript is now acceptable for publication, you may indicate that here to bypass the “Comments to the Author” section, enter your conflict of interest statement in the “Confidential to Editor” section, and submit your "Accept" recommendation.

Reviewer #1: (No Response)

Reviewer #2: All comments have been addressed

2. Is the manuscript technically sound, and do the data support the conclusions?

Reviewer #1: (No Response)

Reviewer #2: Yes

3. Has the statistical analysis been performed appropriately and rigorously? 

Reviewer #1: (No Response)

Reviewer #2: Yes

4. Have the authors made all data underlying the findings in their manuscript fully available?

Reviewer #1: (No Response)

Reviewer #2: Yes

5. Is the manuscript presented in an intelligible fashion and written in standard English?

Reviewer #1: (No Response)

Reviewer #2: Yes

6. Review Comments to the Author

Reviewer #1: The authors have revised the manuscript well. All concerns of the reviewer have been addressed. The manuscript is up to date and can be accepted for publication in current form.

Reviewer #2: The authors carefully address all the concerns of the reviewer, thus, the paper is accepted and recommended for publication.

7. PLOS authors have the option to publish the peer review history of their article (what does this mean?). If published, this will include your full peer review and any attached files.

Reviewer #1: No

Reviewer #2: No

---

## [Editor Report · Acceptance letter]

12 Jan 2024

PONE-D-23-25130R1 

PLOS ONE

Dear Dr. Tran-Huy, 

I'm pleased to inform you that your manuscript has been deemed suitable for publication in PLOS ONE. Congratulations! Your manuscript is now being handed over to our production team.

Kind regards, 

on behalf of

Dr. Musa Hussain 

Academic Editor

PLOS ONE